# Integrated Long-Term Care ‘Neighbourhoods’ to Support Older Populations: Evolving Strategies in Japan and England

**DOI:** 10.3390/ijerph20146352

**Published:** 2023-07-12

**Authors:** Ala Szczepura, Harue Masaki, Deidre Wild, Toshio Nomura, Mark Collinson, Rosie Kneafsey

**Affiliations:** 1Research Centre for Healthcare and Communities, Institute for Health & Wellbeing, Coventry University, Coventry CV1 5FB, UK; deidre_wild@btinternet.com (D.W.); ad6298@coventry.ac.uk (T.N.); aa9398@coventry.ac.uk (R.K.); 2Graduate School of Nursing, Chiba University, Chiba 263-8522, Japan; hmasaki@faculty.chiba-u.jp; 3MC2S Consultancy Services, Bromsgrove, Worcestershire B48 7JX, UK; amarkcollinson@aol.com

**Keywords:** integrated health and social care, national strategy, older people, aging in place, international comparison, Japan, family carers, digital technology, community-based care, COVID-19

## Abstract

Western countries are currently facing the public health challenge of a rapidly aging population and the associated challenge of providing long-term care services to meet its needs with a reduced working age population. As people age, they will increasingly require both health and social care services to maintain their quality of life and these will need to be integrated to provide cost-effective long-term care. The World Health Organization recommended in 2020 that all countries should have integrated long-term care strategies to better support their older populations. Japan, with the most rapidly ageing society in the world, started to address this challenge in the 1990s. In 2017, it introduced a national policy for integrated long-term health and social care services at a local geographical level for older people. England has recently embarked on its first plan aiming for the integration of services for older people. In this article, we compare these approaches to the integration of long-term care systems, including the strengths of each. The paper also considers the effects of historical, cultural and organizational factors and the emerging role of technology. Finally, we identify critical lessons that can inform strategy development in other countries, and highlight the need to provide more international comparisons.

## 1. Introduction

It is estimated that the world population aged 65+ will nearly double from 9.3% in 2020 to 16.0% by 2050 [1]. The fact that people are living longer is a positive reflection on improvements in health and long-term care services; at the same time, falling birth rates are leading to a decrease in the number of working age populations available to provide care [2]. Countries are at different stages in this aging journey. Of these, Japan is the most rapidly aging society, with 28.2% of its population now in this age group [3]. In developed countries, the super-aged population (people aged 85+) poses an additional challenge; the United Kingdom (UK) has the fastest growing super-aged population world-wide, predicted to double by 2041 and treble by 2066 [4]. These two countries therefore offer a useful comparison.

As people grow older, they develop various age-related chronic illnesses, physical disabilities and conditions such as dementia [5]. The integration of health and care services in order to meet these complex care needs is now recognised as a global policy challenge [6]. The World Health Organization (WHO) has recently proposed that all countries should have integrated long-term care strategies to better support their older populations [7]. In this paper, we will focus on strategic macro- or national-level policies designed for the integration of care rather than, as pointed out in a recent systematic review, that adopted by most academic studies to date on the care delivery level or micro-level [8].

## 2. Post-War Development Adult Health and Social Care in Japan and England

Japan established a 10-year strategy to improve long-term care for the elderly (Gold Plan), together with long-term care insurance, in the 1990′s [9], and first introduced a national policy of community-based integrated care in 2017 [10]. In England, the government published its ten-year vision for adult social care reform in 2021 [11], including proposals for the integration of health and social care [12].

Table 1 provides an overview of the national demographic and economic context of long-term health and social care in the two countries. This shows that both countries spend the same proportion of their Gross Domestic Product (GDP) on healthcare, while raising similar amounts of tax revenue relative to this figure. At the same time, Japan has a higher proportion of older people. It also has a higher life expectancy, the highest in the world; this may be considered a positive outcome resulting from national health policies. In contrast to the UK, Japan also faces a halving of its total citizens by 2100, resulting in a significant decrease in the working-age population. While both countries face a similar aging challenge, their different histories have inevitably influenced the integrated long-term care strategy each has developed to better support its aging population.

### 2.1. History of Health and Welfare Policies for Older Adults in Japan

Following the Second World War, the restructuring of healthcare in Japan strengthened community health and public health, and expanded employee-based health insurance and community health insurance [16]. By the 1960′s, universal health insurance coverage had been achieved; this included variable co-payment (public and government) rates that could reach up to 50 per cent for some patients. In 1963, the Social Welfare Services for the Elderly Act led to the creation of care homes and legislation for domiciliary care (see Table 2). However, social care still remained largely dependent on families [17]. Free healthcare for older people was first introduced in 1973. Increased rates of the hospitalization of older adults with predominantly social care needs led to a national ten-year strategy for the Promotion of Health and Welfare for the Elderly (Gold Plan) being introduced in the early 1990s. This was followed by a New Gold Plan and a Long-Term Care Insurance (LTCI) Act to further improve long-term care funding following the economic recession [9]. Changes to family structures [18], a more gender-equal workforce [13], and older people living alone rather than in traditional multi-generation households [19] had inevitably reduced the state’s ability to rely on the unpaid care hitherto provided by family members thereby subsidizing care costs.

The LTCI Act represented a major reform because it introduced insurance specifically for long-term care needs, and also attempted to address the challenge of balancing high-quality care with cost containment [20]. Long-term care was to be funded partly by insurance and partly by co-payment through employer and employee contributions, combined with general taxation [21]. The Act ensured the provision of all necessary care for those aged 65 years and over. Up to that point, citizens were only required to have medical insurance. With the Long-Term Care Act, once a person was aged over 40 years, extra insurance premiums had to be paid. Over time, it became apparent that additional government funding was required to meet the care costs of older adults living in poverty and unable to pay insurance [22]. Additionally, the needs of younger people with disabilities for long-term care were increasingly recognised and a separate national insurance fund was established to cover disabled people aged 40–65 years.

**Table 2 ijerph-20-06352-t002:** Development of Health and Welfare Policies for Older Adults in Japan.

Date: Development	Major Policies
1960s: Beginning of welfare policies for the elderly	1963 Enactment of the Act on Social Welfare Services for the Elderly ◇ Intensive care homes for the elderly created ◇ Legislation on home helpers for the elderly
1970s: Expansion of healthcare expenditures for the elderly	1973 Free healthcare for the elderly
1980s: “Social hospitalization” and “bedridden elderly people” as social problems	1982 Enactment of Health and Medical Services Act for the Aged ◇ Adoption of co-payments for elderly healthcare 1989 Establishment of Gold Plan (10-year strategy for promotion of health and welfare for the elderly) ◇ Promotion of the urgent preparation of facilities and in-home welfare services
1990s: Promotion of the Gold Plan	1994 Establishment of the New Gold Plan (new 10-year strategy for the promotion of health and welfare for the elderly) ◇ Improvement of in-home long-term care
Late 1990s: Preparation for adoption of the Long-Term Care Insurance System	1997 Enactment of the Long-Term Care Insurance Act
2000s: Introduction of the Long-Term Care Insurance System	2000 Enforcement of the Long-Term Care Insurance System

Sources: Ministry of Health Labour and Welfare. Long-Term Care Insurance System of Japan [23].

### 2.2. History of Health and Welfare Policies for Older Adults in England

In England, society’s role in caring for older people was embodied in the 1601 Poor Law, with local parishes having to provide support for their poor. Appendix A: Development of Health and Welfare Policies for Older Adults in England shows that responsibility shifted to local authorities in 1834, who then became responsible for providing accommodation in a workhouse for the poor, with people paying for this through their work. People who were ill could enter an infirmary where care was provided free, but only if they declared themselves a pauper. Following the Second World War, the National Health Service (NHS) was established with free healthcare for all, although general practitioners (GPs) remained as independent clinicians contracted by Health Authorities. Social care was excluded from free health services [24]. Instead, social care for older disabled people was means tested and local authorities (LAs) became responsible for either providing services themselves in a locality or contracting and monitoring other care providers and their service delivery [25]. Over time, an increasing number of older people with long-term healthcare needs were placed in long-stay geriatric wards [26]. These were criticised for having custodial ways of working and providing limited rehabilitation [27]. As a result, in the 1980′s, social care became increasingly concerned with trying to support people ‘in the community’ rather than with institutional care, and improving choice and making care more responsive to individual needs. In 1999, a Royal Commission on Long-Term Care reported deficiencies in the way social care was funded, but the main recommendation to provide free personal care was rejected. This was followed by discussion of a partnership between health and social care with the 2006 White Paper *Our Health, Our Care, Our Say*, seeking the greater integration of health and local authority services for older people. In 2011, degree-level nurse education and specialist geriatric nurse training were introduced [28].

## 3. Resulting Health and Adult Social Care Systems in Japan and England

Japan’s healthcare for older people is largely delivered through hospitals, of which some 80 per cent are owned by the private sector. Both government (national) and private facilities all receive the same not-for-profit reimbursement [29]. Further medical and nursing care is provided in community clinics, health centres and pharmacies [29]. Patients are able to directly access medical specialists, and most Japanese hospital physicians also practice in community clinics. Unlike England, Japanese general practice is still being developed [30,31]. Patients tend to self-diagnose and then consult a specialist physician. If domiciliary care is required, this is largely provided by for-profit, private businesses, but if a nursing home admission is required, costs are covered by insurance and managed by non-profit social welfare corporations [21]. Residential care homes (with no on-site nurses) providing accommodation and 24 h personal care, such as help with washing, dressing, going to the toilet and taking medication are relatively rare in Japan although common in England [32]. Their development has been identified as a future growth area for not-for-profit private providers in Japan [33]. Challenges emerging over time include dissatisfaction with home-based care, poor provision of necessary support for family carers, and fiscal sustainability [34].

In England, NHS hospital and community health services are provided for free, paid for by taxation and accessed primarily via a GP referral. Meanwhile, adult social care services are means tested and referral is via assessment by social and healthcare staff [35]. In general, assessments differentiate between people with healthcare needs (requiring a place in a nursing home) and those who only need ‘care and attention’ (requiring less expensive residential care home admission or domiciliary care provided at home). The initial attempts by government to separate social and nursing care needs in an older person were largely unsuccessful [36], and it has recently been acknowledged that residential homes also provide care for people with complex healthcare needs [37]. Some people living in the community who have a complex social care need (e.g., severe autism) can hold a personal budget to purchase care; this budget can be managed personally or with the help of an independent Care Navigator [38]. People with complex healthcare needs may also qualify for a personal budget while living in the community; if they move into a nursing home, this can subsidise their costs [39]. The resulting pattern of social care provision in England is regularly described as complex, unfair, failing to meet population needs, and poorly understood by the public, who often assume social care to be part of the NHS [40].

## 4. Approaches to Integration of Services for Older People

### 4.1. Evolution of Integrated Community Care ‘Neighbourhood’ System in Japan

In Japan, the economic viability of the New Gold Plan began to face challenges in the new millennium due to the overuse of tests and drugs by doctors and unconstrained demand from patients, resulting in an explosion of costs [41]. In March 2011, the Japan Earthquake highlighted underlying structural problems in the health system that were difficult to resolve fiscally, including those related to long-term care [42]. To help address these, a national policy of devolution to community-based integrated care was introduced for the aging population [10]. This was built upon a model of Health and Welfare Centres that had previously been established in isolated areas, as shown in Figure 1.

At its heart is an Integrated Community Care Support (ICCS) Centre with an expert Care Manager to advise on care plans and help older people keep within a pre-assessed budget allocation, based on insurance designed specifically for long-term care needs introduced by the Long-Term Care Act. Care Managers are expected to ensure nation-wide consistency and fairness, including providing a focus on maintaining health and well-being within culturally supportive communities. They are required to have a national qualification in health, medicine, or welfare, and to have been engaged in work based on that qualification for at least five years. Each ICCS Centre is underpinned by appropriate housing and other forms of support for aging in the community, and a concentration on health promotion and prevention, including the provision of community-based rehabilitation and reablement [44]. The ICCS Centres provide advice on housing and long-term care (excluding nursing homes). The Japanese government’s initial intention was to create a market in health and social care where older people could choose from competitors, including many small community-based care providers. However, large healthcare providers began to expand into care services, which distorted the market [45]. Because Care Managers are mostly funded by such providers, it was considered that this might challenge the perceived independence and fairness of the advice they offer. To deter any conflict of interest, the government levies fines on any provider who puts undue pressure upon a Care Manager. In terms of the care workforce, although there has been some discussion about developing their clinical abilities to support nurses [46], to date, no national strategy has emerged. Conversely, there has been a recent call to increase the number of nurses specializing in Gerontological Nursing and in Home Care Nursing [47].

### 4.2. Evolution of Integrated Community Care System ‘Neighbourhoods’ in England

In England, the world financial crisis of 2007 led to a decade of austerity and a series of policy changes, as shown in Appendix A: Development of Health and Welfare Policies for Older Adults in England. By 2017, only half (49%) of people in care homes were receiving any Local Authority funding [48]. In 2019, the level of government funding for adult social care services fell below 2010/11 levels in real terms, with an extra 1.9 million new clients requesting care [49]. As a result, adult social care services were reported to be at crisis point and on the verge of collapse [50]. In 2019, the government published “The Long-Term Plan”, which proposed the integration of health and social care services at a general population level [51]. It was acknowledged that different histories, cultures, and legal and financial frameworks (including means testing) had hampered the integration of the two services [52]. The Long-Term Plan proposed the establishment of 42 Integrated Care Systems (ICS) across England, each consisting of an integrated care board (with responsibility for spending and performance), and wider integrated care partnerships to address broader population health and social care needs, as shown in Figure 2.

Each ICS structure includes a number of community care ‘neighbourhoods’, each covering a population of 30–50,000 and the associated GP practices. To support these groupings, a new class of “Social Prescribing” link worker is being introduced, and every GP practice in England is expected to have access to a shared link worker by 2023/24. This new role is viewed as a cost-effective way of addressing the fact that one quarter of GP consultations are primarily for a social problem requiring welfare advice, e.g., benefits, employment, housing or debt [54,55]. At present, Social Prescribing Link Workers can only ‘prescribe’ services provided free by charities or voluntary organizations [56,57]. These posts are also not specifically dedicated to the needs of older people.

A further government White Paper “Working together to improve health and social care for all” published in February 2021 signalled a move away from internal market structures and towards integrated care structures for older people [58]. An OECD critique of the marketisation of long-term residential and nursing home care had highlighted providers that were unable to respond to competitive forces without compromising care quality [59]. This White Paper was followed in September 2021 by a post-pandemic recovery plan “Build Back Better: Our Plan for Health and Social Care”, which introduced a new nationwide health and social levy designed to provide extra funding for social care and the NHS [60]. The plan also raised the personal resource threshold (means-testing) for access to LA-funded care on a sliding scale, although it did not address the ‘divide’ between free healthcare for conditions like cancer and personal contributions to social care for dementia [61]. It also appeared that the recovery plan might do little ‘to improve quality of care’ [62], with questions raised about its effective integration [63]. In December 2021, a ten-year vision for adult social care reform was presented in the White Paper “People at the Heart of Care: Adult Social Care Reform” [11]. This was rapidly followed by a further White Paper “Joining up care for people, places and populations” published in February 2022, with proposals for health and care integration [12]. Although this latest document outlined a broad framework, it still contained limited detail [53]. In July 2022, a new Health and Care Act placed this restructuring on a statutory footing [64].

## 5. Discussion

In Japan, once it became clear in the 1990′s that an aging population would increase costs to an unacceptable level, the government was proactive in introducing long-term national funding reforms for older people’s care. Cross-party consultations were held at the government level to agree upon the long-term policy reforms required. The initial reforms (Gold Plan/New Gold Plan) were followed in 2017 by a national policy of community-based integrated care for older people, as shown in Table 2. In the UK, with a lack of cross-party consultation, governments have tended to focus on short-term initiatives until recently, as shown in Appendix A [65,66]. Any change needs to be managed within a complex adult social care sector with a mix of private for-profit, charitable and local authority providers [48]. There are also deep-seated structural issues with many larger providers owned by private equity funds abroad, attracted to the market due to its potential for refinancing [67]. To date, there has been little consensus on how budgetary control can be equitably managed across both health and care sectors [68,69].

Although the 42 area-based English ICSs are based on geographical localities [51], similar to Japan’s ICCS Centres, there is no exclusive focus on integrated long-term care and considerable individual flexibility, which may lead to geographical variations; for example, the needs of older people may depend not only on their age and other personal demographic factors, but also on whether they live in an urban or rural community. This type of “post code lottery” has recently been reported regarding the provision of domiciliary or home-care services across England [70]. It is also recognised that guidance is not offered to individuals early in their aging care journey, unlike Japan, with choices about care options often made in a crisis situation, for example, when an older person is being discharged from hospital [56]. In such situations, individuals and their families report a lack of information, limited engagement, and feeling powerless to control the situation [71]. This compares to Japan, where older people have access to the support of an expert Care Manager early in the aging process to help them optimise and personalise decisions about their health/social care and housing needs as these emerge and how to use their allocated budget. 

In Japan, the ICCS Care Manager is able to refer people from a locality to appropriate housing, support for living, community-based rehabilitation and reablement, with an emphasis on ensuring national consistency. England has no directly comparable workforce. Social Prescribing Link Workers have a more limited role; they cannot refer to a similarly wide range of support services, and they do not focus exclusively on the older population [72]. There is no requirement for a national qualification or for relevant experience for a broader role, with currently ‘no set entry requirements’ [73]. However, through their attachment to a general practice team, they will have access to GP support if required [56]. In contrast, ICCS Care Managers cannot easily access GP support because primary care is still being established in Japan [30]. In England, historical attempts to establish a national cadre of GPs with a special interest in older people have generated limited interest [74,75]. Strategies are now being piloted in some areas to integrate GP services directly into care homes [76]. An expanded role for community geriatricians is not included in the latest proposals for health and care integration [12], even though these physicians have the expertise to support the integration of health and care for older people [68,77].

Both countries face falling birth-rates and a future shortage of care workers to support their aging populations, highlighting a possible need to recruit overseas staff to fill vacancies [78]. Japan has, to date, remained reliant on the country’s own population. This is partly because of language barriers, but also due to historical concerns that non-indigenous staff might be less qualified or dilute the ethos of caring professions [79]. In England, this route is well established with nearly one in five current care workers born outside the UK [80]. Many of these migrant workers report that they are overqualified for their current roles [81]. Post-Brexit immigration rules may restrict the supply of foreign care workers [82].

In Japan, society’s positive attitude towards those in later life [83] has been utilised to build a national system, with a focus on the use of volunteers to complement the work of trained professionals [84]. In England, a less positive view of older people and widespread ageism has been reported in society [85,86]. However, during the COVID-19 pandemic, the national mobilization of volunteers was successful in supporting older people. Although the ten-year vision for adult social care includes building volunteering capacity [11], it has been acknowledged that this may be difficult [87]. In Japan, there is a continued reliance upon unpaid care provided by family members to subsidise care costs; England does not have a well thought through strategy to integrate the large body of unpaid informal carers [88]. 

In England, it is unclear how, and to what extent, care homes will be incorporated into the new ICS structures [62,63,89]. The pandemic catalysed closer working between the NHS and care home staff, enabling an enhanced role to be imagined for the social care staff [90]. An “enhanced health in care homes” framework has been updated [91] to include new areas of expertise to be developed by social care workers [92,93]. Attention is now being paid to enhanced roles for nurses employed in nursing homes, with calls for more trained specialist gerontological nurses and a clearer role for their contribution to the integration of health and social care [94]. In Japan, there have been similar calls to increase the numbers of nurses specializing in gerontological nursing [47]. Some early discussion focused on developing the clinical skills of support staff in nursing homes [46], but little attention has been paid to upskilling social care workers. To date, Japan has no national strategy in either area.

### Future Innovation and Transformation

As well as structural changes, there are moves in both countries to ‘transform’ care by using technology to enable care to be provided in fundamentally new ways. In Japan, ‘Care Science’ has been designated as a new discipline to complement medical science and nursing science, to accelerate the development of technologies such as assistive robots, sensors and artificial intelligence (AI) [95]. In England, the ten-year vision has highlighted the important role of innovation in adult social care, including in care homes [11]. Evidence suggests that innovations are unlikely to be adopted by English care homes unless benefits can be demonstrated [96]. To address this, a ‘living lab’ approach has been successfully piloted to evaluate and demonstrate similar innovations *in situ* including sensors, robotics, and digital information systems [97]. The latter will require national training to upskill a workforce that may not be digitally competent [98]. 

In 2021, Japan’s Ministry of Health established a long-term care information system (LIFE) as part of longer term care fee revisions [99]. One aim is to support Care Science by promoting new ways of working through digital transformation (DX) so that sustainability can be ensured for care systems. In England, the Care Quality Commission, which regulates healthcare and social care, is developing a new strategy to monitor care and improve care quality using digital data collection [100]. A recent agreement with the National Institute for Health and Care Excellence (NICE) aims to improve the use of data and information (including AI and digital health technologies) to transform care [101]. NICE began to establish an evidence collection similar to LIFE, but, to date, this contains relatively few items on social care [102]. In June 2022, NHS England published its “Plan for digital health and social care” [103]. This includes digitizing health and social care records in all 42 ICSs (with 80% of registered social care providers to have digital care records by March 2024) and broadband upgrades to allow remote support and technology-enabled care in all care homes.

## 6. Conclusions

The analysis presented here compares policy development in two countries with different historical, cultural and health and social care backgrounds. In Japan, the adult health and social care system is newer and less complicated than in England, both in terms of its not-for-profit structures and its age-based insurance funding, and integration is more advanced. It could be argued that this context is more amenable to reform than England, where separate, independently funded services have evolved over several centuries. Recent political changes may slow this process even further. In Japan, the introduction of personal care budgets, the creation of certified Care Managers to provide specific advice, and the fact that government providers and private facilities receive the same not-for-profit national payment rates all help ensure national consistency and fairness. In England, the newly established ICSs, which are not specifically focused on the aging population, are able to develop local strategies, which may increase the likelihood of regional variation and perceived unfairness. In both countries, policies to support future innovation and technology diffusion in the care sector are developing rapidly with implementation that is, possibly, less well developed at a national level in Japan. It is recommended that common definitions of terms should now be developed, including for ‘Care Science’. Both countries are attempting to address issues of sustainability and, for this, there may be critical lessons to be drawn for other countries, as recommended by the WHO [7]. Other countries at different stages of implementing their integrated long-term health and care policies may also find this paper of value. This includes, for example, the United States, where national and state policies developed over the last ten years are now being re-examined [104], or China, where integration is currently being considered within a new policy landscape [105].

## Figures and Tables

**Figure 1 ijerph-20-06352-f001:**
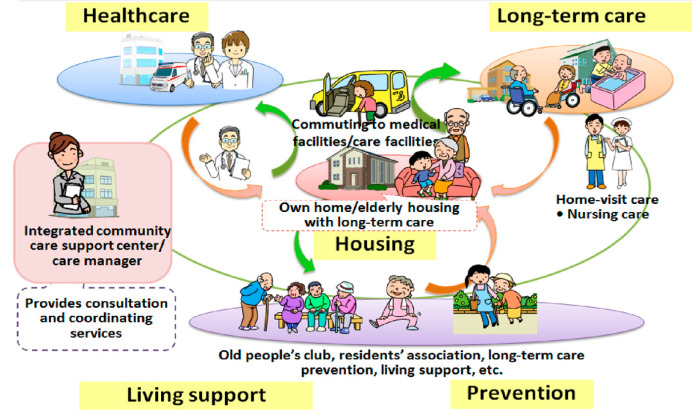
Integrated community care support ‘neighbourhood’ system in Japan. Source: Ministry of Health, Labour and Welfare [43].

**Figure 2 ijerph-20-06352-f002:**
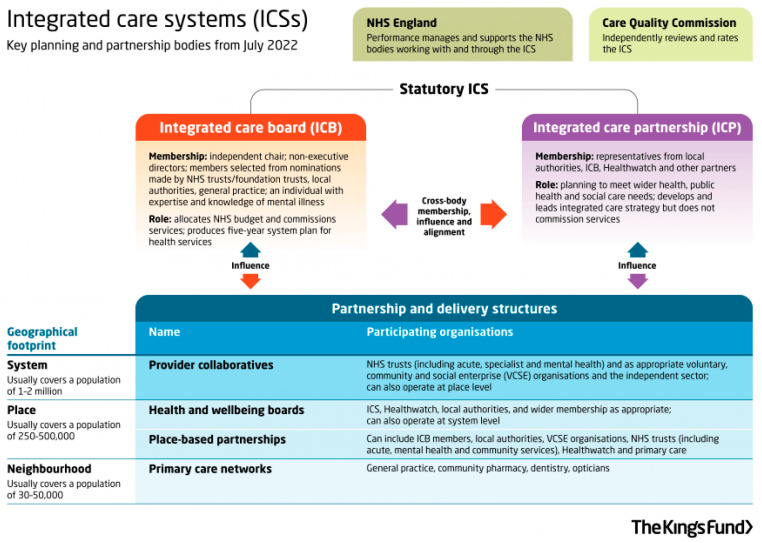
Integrated community care system ‘neighbourhood’ in England (key planning and partnership bodies). Source: The King’s Fund 2022 [53].

**Table 1 ijerph-20-06352-t001:** Comparison of Long-Term Care Challenges in Japan and UK.

Comparator	Japan	UK
Total population	127.4 million	66.5 million
Mean life expectancy	83.9 years	81.1 years
Percentage of population aged 65 and over in 2019	27.3	16.0
Percentage of population predicted to be aged ≥65 by 2040	34.2	24.3
Percentage of GDP spent on health	9.4	9.4
Tax revenue as percentage of GDP	32.0	32.9
Predicted total population in 2100	59.7 million	71.4 million

Sources: [1,13,14,15].

## Data Availability

No new data were created or analyzed in this study. Data sharing is not applicable to this article.

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
