# Peer review of "Integrated Long-Term Care ‘Neighbourhoods’ to Support Older Populations: Evolving Strategies in Japan and England"

_ijerph, 2023, doi:10.3390/ijerph20146352_

Round 1

Reviewer 1 Report

General Comments

The abstract is good. The article is good. Rather than reporting on research, the article offers a comparative analysis of older adult health and social care efforts and challenges in Japan and England. The treatment of the subject is very good in terms of references and cited differences between the two countries ranging from policy to technological innovations. I believe the article is of strong interest to a wide range of readers, and that it offers unique and important insights that other countries may benefit from exploring in the context of health and social service systems and the care of older adult populations.

This said, I believe the article is publishable after consideration to some areas of revision:

1.      The last sentence in the introduction seems to be a purpose statement, but it could be more clearly articulated as the purpose of the article.

2.      My first major concern relates to the orientation of the article as evidenced in the background literature and interestingly also in some of the conclusions. The authors seem to hold an ‘apocalyptic demography’ viewpoint, ‘catastrophizing’ demographic aging and implying that all older adults are heavy users of health system resources. In the context of their broader interest on the social determinants of health, and the importance of considering integrated and coordinated health AND social care for meeting the needs of older adults in a health and quality of life sense, I was surprised to see this. The authors should find additional data and references that reflect who the older users of health and social care systems in England and Japan are. This is important also in helping to not perpetuate ageist views especially when the reality is that only a small proportion of older adults are high consumers of health services, often only in the last years of life. The authors also might reference the types of social care resources that are most important to older adults. It is indeed true that the need for adequate and appropriate health and social care resources will grow because the proportions of older adults in the population are growing, but it is inaccurate to imply that all older adults are high consumers of such services.

3.      To help to reduce ageism and support equity, diversity and inclusion principles in Figure 1 the authors should refrain from using the terms ‘elderly’ and ‘old people’ and use ‘older adults’ instead and consider more inclusive images and roles in the graphic representations that are depicted.

4.      The authors suggest that local solutions made by Integrated Care Systems in England are a threat to equality in service provision. This is part of the equity versus equality argument, and they should clarify and be careful about what they mean. If health profiles are different across different geographies, it is important to be responsive to those local needs in order to best promote and support health and wellness rather than with egalitarian cookie cutter approaches that assume everybody everywhere is the same with the same needs. A key, and well referenced example of this is the differences between the needs and populations living in urban and rural areas in terms of age, gender, education, income, chronic conditions, access to care etc. Urban solutions don’t often translate well into rural communities and the reverse is also true.

5.      In the discussion section, I think a table summarizing some of the differences between England and Japan on some metrics could be helpful.

6.      In the conclusions it might be better to reference evidence from China and US in a future directions section rather than at the outset of this section.

Minor editorials

-in section 2.2 LAs are not defined

-in Section 3.1 ‘residential care homes’ should be defined

-in Section 4. 1

               -could the authors more fully explain the personal care budgets that are part of Japanese policy?

               -In Figure 1         – don’t use ‘old people’s club’ or ‘elderly’

                                             - does the colour of the arrows mean anything?

-maybe the ICCS should be brought from the outside of the figure to the centre beside housing

-what does re-enablement mean?

-page 6 lines 202-203 specify some of the social problems that people go to GPs for

Author Response

The abstract is good. The article is good. Rather than reporting on research, the article offers a
comparative analysis of older adult health and social care efforts and challenges in Japan and
England. The treatment of the subject is very good in terms of references and cited differences
between the two countries ranging from policy to technological innovations. I believe the article is of
strong interest to a wide range of readers, and that it offers unique and important insights that other
countries may benefit from exploring in the context of health and social service systems and the care
of older adult populations. This said, I believe the article is publishable after consideration to some
areas of revision
Response:
We thank the reviewer for this very positive overview.
__________________________________________________________________________________
Comment 1:
The last sentence in the introduction seems to be a purpose statement, but it could be more clearly
articulated as the purpose of the article

Response 1:
The final sentence has been changed from: “At the same time, a systematic review of the evidence to
date has identified that academic studies of integration largely focus on the care delivery level (microlevel)
rather than more strategic macro- or national-level policies (8).”
To: “In this paper we will focus on strategic macro- or national-level policies for integration of care
rather than, as pointed out in a recent systematic review, on that adopted by most academic studies
to date on the care delivery level or micro-level [8]”
__________________________________________________________________________________
Comment 2:
My first major concern relates to the orientation of the article as evidenced in the background
literature and interestingly also in some of the conclusions. The authors seem to hold an ‘apocalyptic
demography’ viewpoint, ‘catastrophizing’ demographic aging and implying that all older adults are
heavy users of health system resources. In the context of their broader interest on the social
determinants of health, and the importance of considering integrated and coordinated health AND
social care for meeting the needs of older adults in a health and quality of life sense, I was surprised
to see this. The authors should find additional data and references that reflect who the older users of
health and social care systems in England and Japan are. This is important also in helping to not
perpetuate ageist views especially when the reality is that only a small proportion of older adults are
high consumers of health services, often only in the last years of life. The authors also might
reference the types of social care resources that are most important to older adults. It is indeed true
that the need for adequate and appropriate health and social care resources will grow because the
proportions of older adults in the population are growing, but it is inaccurate to imply that all older
adults are high consumers of such services.
Response 2:
We do not consider that we catastrophise – but there are real challenges in terms of longer lives as
shown in the figure below for healthcare costs alone, and these are multiplied for social care costs.
The reviewer her/himself identifies ”need for adequate and appropriate health and social care
resources will grow because the proportions of older adults in the population are growing “. We point
out that the issue is that older people are larger consumers of healthcare & social care while no longer
contributing significantly to the costs of these in the Discussion. The underlying reduced birth rate was
already considered in the Discussion (see lines 293 onwards), and is now also highlighted in earlier
insertion in lines 33-34.

Comment 3:
To help to reduce ageism and support equity, diversity and inclusion principles in Figure 1 the authors
should refrain from using the terms ‘elderly’ and ‘old people’ and use ‘older adults’ instead and
consider more inclusive images and roles in the graphic representations that are depicted
Response 3:
Please see response below in Table Minor Comments: This figure is taken from a Japanese
government document – so we are unable to alter it.
__________________________________________________________________________________
Comment 4:
The authors suggest that local solutions made by Integrated Care Systems in England are a threat to
equality in service provision. This is part of the equity versus equality argument, and they should
clarify and be careful about what they mean. If health profiles are different across different
geographies, it is important to be responsive to those local needs in order to best promote and
support health and wellness rather than with egalitarian cookie cutter approaches that assume
everybody everywhere is the same with the same needs. A key, and well referenced example of this is
the differences between the needs and populations living in urban and rural areas in terms of age,
gender, education, income, chronic conditions, access to care etc. Urban solutions don’t often
translate well into rural communities and the reverse is also true
Response 4:
We are well versed in the issue of equality versus equity. We mentioned this in terms of ‘postcode
lottery’ in the Discussion (see line 268). Because this phrase may not be understood by all your
readers, we now explain in lines 265-267 in the previous sentence “for example, the needs of older
people may depend not only on their age and other personal demographic factors, but also on
whether they live in an urban or rural community”.
__________________________________________________________________________________
Comment 5:
In the discussion section, I think a table summarizing some of the differences between England and
Japan on some metrics could be helpful
Response 5:
In Table 1 we have summarised some differences between the two countries in terms of various
metrics relevant to sustainable long-term care provision.
Reviewer 2 also suggests (see Comment 4) that we “produce a second separate table presenting the
development and chronology of events in England’s long-term care and social welfare policy” to
compare with the one for Japan. We have produced a similar table for England but this is far longer
than Table 1. We have therefore produced this as supplementary material, Table S1: Development of
Social and Healthcare Policies for Older Adults in England, referred to in the text at two places lines
105-106 and lines 202-203.
Rather than placing this further table in the Discussion, as suggested by Reviewer 1, we have
adopted Reviewer 2’s suggestion and now refer to the two tables in the Discussion (lines 255-257);
we have also replaced the previous ref 65 with the two references cited as sources for Table S1.
__________________________________________________________________________________

Comment 6:
In the conclusions it might be better to reference evidence from China and US in a future directions
section rather than at the outset of this section
Response 6:
We thank the reviewer for this advice. We agree that these sentences do not fit well at the start of
the Conclusions section. However, we have instead moved them to the very end, where we consider
they fit well.
__________________________________________________________________________________
Table: Minor Comments:
Section 2.2 LAs are not defined Now expanded definition
Section 3.1 ‘residential care homes’ should be defined Now defined lines
Section 4. 1 could the authors more fully explain the personal
care budgets that are part of Japanese policy?
We now expand on this i.e. based on
insurance specifically for long-term
care needs introduced by the Long-
Term Care Act
Fig 1: a) don’t use ‘old people’s club’ or ‘elderly’ This figure is taken from the Japanese
government document cited – so we
are unable to alter it. Apologies.
Arrow colours have no meaning.
Fig 1: b) does the colour of the arrows mean anything?
Fig 1: c) maybe the ICCS should be brought from the outside
of the figure to the centre beside housing
(Page 5, Line 165) what does re-enablement mean? Corrected- should be ‘reablement’
Page 6 lines 202-203 specify some of the social problems
that people go to GPs for
Details of welfare problems now
added.

Reviewer 2 Report

This manuscript compares the policy and implementation of long-term care and social welfare systems in Japan and the United Kingdom, with a particular focus on how the concept of “Neighbourhoods” supports the care needs of the elderly in the community. This emphasis can provide readers with comparative information about Eastern and Western countries, which is meaningful. Several comments and suggestions as below for this manuscript.

1.     This manuscript advocates exploring the national policy for integrated long-term health and social care services of elderly care at the national level, choosing Japan and England for comparison, but it is suggested to explain the particularity of choosing these two countries. In addition, explain the sources of empirical data used to compare the data of the two countries and the methods used to compare the empirical data.

2.     Although the authors point out that both countries face similar aging challenges, the data in Table 1 show that Japan has a higher percentage of older people than the UK. And Japanese elderly people live longer. The authors can further analyze the significance of the differences in national policies and budgetary expenditures based on the care needs of the elderly.

3.     It is recommended that the authors further analyze the core elements of the Japanese government's promotion of community-integrated care systems, especially the impact of " integration of local resources" and "community mutual assistance" on the community mutual aid function. And compare this part of Japan with Western countries, such as the community where the elderly live in the United Kingdom, in the operation of the community-integrated care system.

4.     It is recommended that the authors present a separate table presenting the development and chronology of events in England’s long-term care and social welfare policy for the elderly after World War II. And analyze the England's policy of "integrating local resources" and "community mutual assistance" and its actual impact on the integrated long-term care of the elderly.

5.     In this manuscript, the term "neighborhoods" is proposed for the use of community care in the UK's ICS framework, but the service name "neighborhoods" is not seen for the method used by Japan's LTCI. It is suggested that the names of long-term care and social welfare resources for the elderly in the communities of the two countries should use close nouns and give clear definitions.

6.     In the Discussion: Japan's long-term care insurance system also introduces a market mechanism to allow the insured to purchase services themselves to strengthen self-selection. Moreover, Japan's long-term care service market is also highly commoditized. Therefore, how to successfully develop the concept of “ Neighbourhoods” in the community to provide the long-term care needs of the elderly is the part that needs to be discussed in depth in this article.

7.     In the Discussion: Countries around the world are facing a crisis of insufficient care manpower. It is suggested that this manuscript discusses how social mutual assistance mechanisms can be developed to strengthen the community life care and support of elderly people. In addition, discuss Japan's comprehensive community care system, and how to make use of the experience of community residents or voluntary groups to cooperate with each other to provide services.

The article is written in English and can be read.

Author Response

Comment 1:
This manuscript advocates exploring the national policy for integrated long-term health and social
care services of elderly care at the national level, choosing Japan and England for comparison, but it
is suggested to explain the particularity of choosing these two countries. In addition, explain the
sources of empirical data used to compare the data of the two countries and the methods used to
compare the empirical data.
Response 1:
The reason for choosing these two countries is explained in the first paragraph. Japanese society has
aged most rapidly worldwide with 28.2% of the population now aged over 65+ years & UK society
has the fastest growing super-aged (85+ years) population worldwide. This is further clarified with
an additional sentence at the end of this paragraph “These two countries therefore offer a useful
comparison”. Empirical data is fully referenced throughout.
__________________________________________________________________________________
Comment 2:
Although the authors point out that both countries face similar aging challenges, the data in Table 1
show that Japan has a higher percentage of older people than the UK. And Japanese elderly people
live longer. The authors can further analyze the significance of the differences in national policies and
budgetary expenditures based on the care needs of the elderly

Response 2:
We did explain this in the text associated with Table 1. The fact that people in Japan live longer is a
success, but the falling birth rate represents a challenge since not only is a workforce needed to
provide care, but also to pay taxes to fund care workers. This is not the main focus of the current
article, but we acknowledge that Japan also faces a halving of its total citizens by 2100, resulting in a
significant decrease in the working age population This is also addressed as part of our response to
Comment 2 by Reviewer 1.
__________________________________________________________________________________
Comment 3:
It is recommended that the authors further analyze the core elements of the Japanese government's
promotion of community-integrated care systems, especially the impact of " integration of local
resources" and "community mutual assistance" on the community mutual aid function. And compare
this part of Japan with Western countries, such as the community where the elderly live in the United
Kingdom, in the operation of the community-integrated care system
Response 3:
The united Kingdom does not yet have evidence on the operation of the community-integrated care
system, so we are unable to analyse this as suggested. However, we now provide a new table on the
development of the community-integrated care system in England (Table S1: Development of Social
and Healthcare Policies for Older Adults in England). We now also detail some of the policies that have
led to community-integrated care in section 2.2 (lines 118-127).
__________________________________________________________________________________
Comment 4:
It is recommended that the authors present a separate table presenting the development and
chronology of events in England’s long-term care and social welfare policy for the elderly after World
War II. And analyze the England's policy of "integrating local resources" and "community mutual
assistance" and its actual impact on the integrated long-term care of the elderly.
Response 4:
We now provide a separate table on the development of long-term care and social welfare policy in
England. Because this table is longer than the one for Japan, it is provided as supplementary material
as Table S1: Development of Social and Healthcare Policies for Older Adults in England.
The table includes development of England’s policy that addresses "integrating local resources" and
"community mutual assistance".
__________________________________________________________________________________
Comment 5:
In this manuscript, the term "neighborhoods" is proposed for the use of community care in the UK's
ICS framework, but the service name "neighborhoods" is not seen for the method used by Japan's
LTCI. It is suggested that the names of long-term care and social welfare resources for the elderly in
the communities of the two countries should use close nouns and give clear definitions.
Response 5:
We welcome the suggestion that there is a need for clear definitions of terminologies used in the two
countries. We now include this as a recommendation “It is recommended that common definitions
of terms should now be developed, including for ‘Care Science’.” (see line 369-371).

__________________________________________________________________________________
Comment 6:
In the Discussion: Japan's long-term care insurance system also introduces a market mechanism to
allow the insured to purchase services themselves to strengthen self-selection. Moreover, Japan's
long-term care service market is also highly commoditized. Therefore, how to successfully develop the
concept of “ Neighbourhoods” in the community to provide the long-term care needs of the elderly is
the part that needs to be discussed in depth in this article
Response 6:
This is an interesting point. We agree that this is important, but we consider that it cannot be discussed
in depth in this article. This would require an entirely separate article to do the topic justice. We do
now, however, allude to recent discussion on future funding in England in response to Comment 4
from Reviewer 1.
__________________________________________________________________________________
Comment 7:
In the Discussion: Countries around the world are facing a crisis of insufficient care manpower. It is
suggested that this manuscript discusses how social mutual assistance mechanisms can be developed
to strengthen the community life care and support of elderly people. In addition, discuss Japan's
comprehensive community care system, and how to make use of the experience of community
residents or voluntary groups to cooperate with each other to provide services.
Response 7:
We thank the reviewer for this summary of our article.
__________________________________________________________________________________
We would like to thank both reviewers for their useful comments and suggestions, and we look
forward to receiving a final decision on our revised manuscript.